# AdaFlow: Imitation Learning with Variance-Adaptive Flow-Based Policies

**Xixi Hu, Bo Liu, Xingchao Liu, Qiang Liu**
The University of Texas at Austin
{hxixi,bliu,xcliu,lqiang}@cs.utexas.edu

## Abstract

Diffusion-based imitation learning improves Behavioral Cloning (BC) on multi-modal decision-making, but comes at the cost of significantly slower inference due to the recursion in the diffusion process. It urges us to design efficient policy generators while keeping the ability to generate diverse actions. To address this challenge, we propose AdaFlow, an imitation learning framework based on flow-based generative modeling. AdaFlow represents the policy with state-conditioned ordinary differential equations (ODEs), which are known as probability flows. We reveal an intriguing connection between the conditional variance of their training loss and the discretization error of the ODEs. With this insight, we propose a variance-adaptive ODE solver that can adjust its step size in the inference stage, making AdaFlow an adaptive decision-maker, offering rapid inference without sacrificing diversity. Interestingly, it automatically reduces to a one-step generator when the action distribution is uni-modal. Our comprehensive empirical evaluation shows that AdaFlow achieves high performance with fast inference speed.

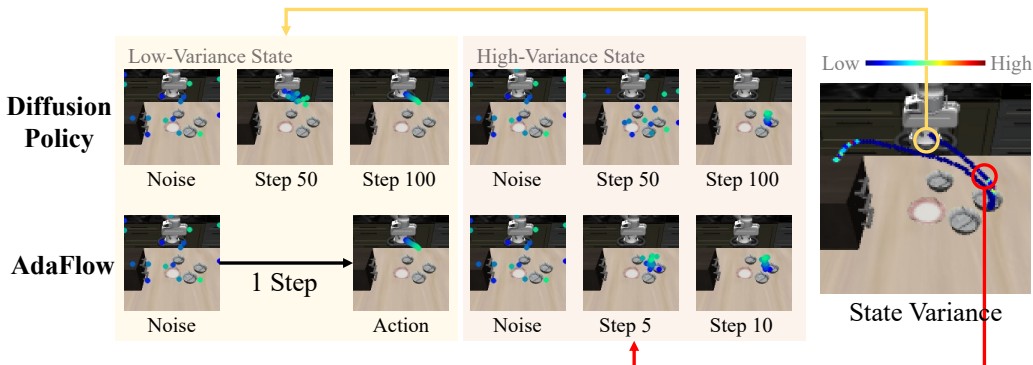

Figure 1: AdaFlow is a fast imitation learning policy. It adaptively adjust the number of simulation steps when generating actions. For low-variance states, it functions as a one-step action generator. For high-variance states, it employs more steps to ensure accurate action generation. This adaptive approach enables AdaFlow to achieve an average generation speed close to one step per task completion.

## 1 Introduction

Imitation Learning (IL) is a widely adopted method in robot learning [1, 2]. In IL, an agent is given a demonstration dataset from a human expert finishing a certain task, and the goal is for it to complete the task by learning from this dataset. IL is notably effective for learning complex, non-declarative motions, yielding remarkable successes in training real robots [3–6].

38th Conference on Neural Information Processing Systems (NeurIPS 2024).

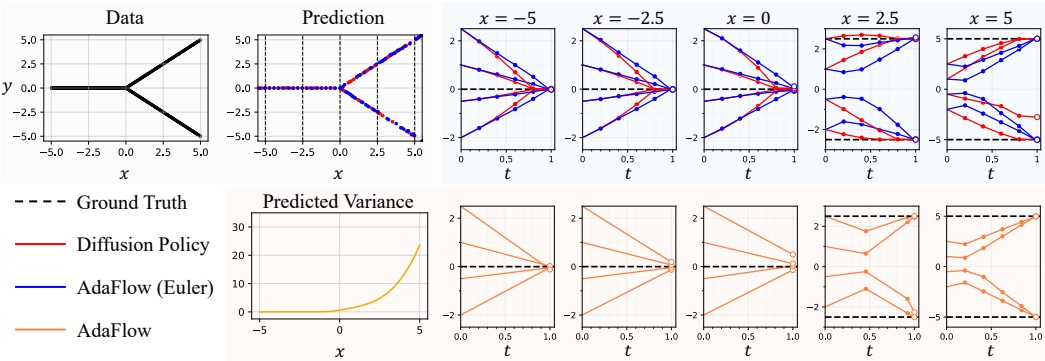

Figure 2: Illustrating the computation adaptivity of AdaFlow (orange) on simple regression task. In the upper portion of the image, we use Diffusion Policy (DDIM) and AdaFlow to predict $y$ given $x$, with deterministic $y = 0$ when $x \leq 0$, and bimodal $y = \pm x$ when $x > 0$. Both DDIM and AdaFlow fit the demonstration data well. However, the simulated ODE trajectory learned by Diffusion-Policy with DDIM (red) is not straight no matter what $x$ is. By contrast, the simulated ODE trajectory learned by AdaFlow with fixed step (blue) is a straight line when the prediction is deterministic ($x \leq 0$), which means the generation can be exactly done by one-step Euler discretization. At the bottom, we show that AdaFlow can adaptively adjust the number of simulation steps based on the $x$ value according to the estimated variance at $x$.

The primary approach for IL is Behavioral Cloning (BC) [7–10], where the agent is trained with supervised learning to acquire a deterministic mapping from states to actions. Despite its simplicity, vanilla BC struggles to learn diverse behaviors in states with many possible actions [10, 11, 6, 12]. To improve it, various frameworks have been proposed. For instance, Implicit BC [12] learns an energy-based model for each state and searches the actions that minimize the energy with optimization algorithms. Diffuser [13, 14] and Diffusion Policy [6] adopts diffusion models [15, 16] to generate diverse actions, which has become the default method for training on large-scale robotics data [17–20].

The computational cost of the learned policies at the execution stage is important for an IL framework in a real-world deployment. Unfortunately, none of the previous frameworks enjoys both efficient inference and diversity. Although energy-based models and diffusion models can generate multi-modal action distributions, they require *recursive processes* to generate the actions. These recursive processes usually involve tens (or even hundreds) of queries before reaching their stopping criteria.

In this paper, we propose a new IL framework, named AdaFlow, that learns a dynamic generative policy that can autonomously adjust its computation on the fly, thus cheaply outputs multi-modal action distributions to complete the task. AdaFlow is inspired by recent advancements in flow-based generative modeling [21–24]. We learn probability flows, which are essentially ordinary differential equations (ODEs), to represent the policies. These flows are powerful generative models that precisely capture the complicated distributions, but similar to energy-based models and diffusion models, they still require *multiple recursive iterations* to simulate the ODEs in the inference stage.

AdaFlow differs from existing flow generative models like Rectified Flow [25] and Consistency Models [26], by utilizing the *initially learned ODE* to maintain low training and inference costs, and function as a one-step generator for deterministic target distributions. In contrast, both of these methods require an additional distillation or reflow [25] process to achieve fast inference. To improve the efficiency, we propose an adaptive ODE solver based on the finding that the simulation error of the ODE is closely related to the variance of the training loss at different states. We let the action generation model to output an additional variance scalar alongside the action it produces. During the execution of the policy, we change the step size according to the variance predicted at the current state. Equipping the flow-based policy with the proposed adaptive ODE solver, AdaFlow wisely allocates computational resources, yielding high efficiency without sacrificing the diversity provided by the flow-based generative models. Specifically, in states with deterministic action distributions, AdaFlow generates the action in *one step* – as efficient as naive BC.

Empirical results across multiple benchmarks demonstrate that AdaFlow achieve consistently good performance across success rate with high execution efficiency. Specifically, our contributions are:

- We proposed AdaFlow as a generative model-based policy for decision making tasks, capable of generating actions almost as quickly as a single model inference pass.

- We conducted comprehensive experiments across decision making tasks, including navigation and robot manipulation, utilizing benchmarks such as LIBERO [27] and RoboMimic [10]. AdaFlow consistently outperforms existing state-of-the-art models, despite requiring 10x less inference times to generate actions.

- We offer a theoretical analysis of the overall error in action generation by AdaFlow, providing a bound that ensures precision and reliability.

## 2 Related Work

**Diffusion/Flow-based Generative Models and Adaptive Inference**    Diffusion models [28, 16, 15, 29] succeed in various applications, e.g., image/video generation [30–33], audio generation [34], point cloud generation [35–38], etc.. However, numerical simulation of the diffusion processes typically involve hundreds of steps, resulting in high inference cost. Post-hoc samplers have been proposed to solve this issue [39–44] by transforming the diffusion process into marginal-preserving probability flow ODEs, yet they still use the same number of inference steps for different states. Although adaptive ODE solvers, such as adaptive step-size Runge-Kutta [45], exist, they cannot significantly reduce the number of inference steps. In comparison, the adaptive sampling strategy of AdaFlow is specifically designed based on intrinsic properties of the ODE learned rectified flow, and can achieve one-step simulation for most of the states, making it much faster for decision-making tasks in real-world applications. Recently, new generative models [21, 25, 22, 23, 46, 24, 26, 47] have emerged. These models directly learn probability flow ODEs by constructing linear interpolations between two distributions, or learn to distill a pretrained diffusion model [26, 47] with an additional distillation training phase. Empirically, these methods exhibit more efficient inference due to their preference of straight trajectories. Among them, Rectified flow achieves one step generation with reflow, a process to straighten the ODE. However, it requires a costly synthetic data construction.

By contrast, AdaFlow only leverages the initially learned ODE, but still keeps cheap training and inference costs that are similar to behavior cloning. We achieve this by unveiling a previously overlooked feature of these flow-based generative models: they act as one-step generators for deterministic target distributions, and their variance indicates the straightness of the probability flows for a certain state. Leveraging this feature, we design AdaFlow to automatically change the level of action modalities given on the states.

**Diffusion Models for Decision Making**    For decision making, diffusion models obtain success as in other applications areas [48–51]. In a pioneering work, Janner et al. [13] proposed "Diffuser", a planning algorithm with diffusion models for offline reinforcement learning. This framework is extended to other tasks in the context of *offline reinforcement learning* [52], where the training dataset includes reward values. For example, Ajay et al. [14] propose to model policies as conditional diffusion models. The application of DDPM [16] and DDIM [43] on visuomotor policy learning for physical robots [6] outperforms counterparts like Behavioral Cloning. Freund et al. [53] exploits two coupled normalizing flows to learn the distribution of expert states, and use that as a reward to train an RL agent for imitation learning. AdaFlow admits a much simpler training and inference pipeline compared with it. Despite the success of adopting generative diffusion models as decision makers in previous works, they also bring redundant computation, limiting their application in real-time, low-latency decision-making scenarios for autonomous robots. AdaFlow propose to leverage rectified flow instead of diffusion models, facilitating adaptive decision making for different states while significantly reducing computational requirements. In this work, similar to Diffusion Policy [6], we focus on offline imitation learning. While AdaFlow could theoretically be adapted for offline reinforcement learning, we leave it for future works.

## 3 AdaFlow for Imitation Learning

To yield an agent that enjoys both multi-modal decision-making and fast execution, we propose AdaFlow, an imitation learning framework based on flow-based generative policy. The merits of AdaFlow lie in its adaptive ability: it identifies the behavioral complexity at a state before allocating

computation. If the state has a deterministic choice of action, it outputs the required action rapidly; otherwise, it spends more inference time to take full advantage of the flow-based generative policy. This handy adaptivity is made possible by leveraging a combination of elements: 1) a special property of the flow 2) a variance estimation neural network and 3) a variance-adaptive ODE solver. We formally introduce the whole framework in the sequel.

## 3.1 Flow-Based Generative Policy

Given the expert dataset $D = \left\{ (s^{(i)}, a^{(i)}) \right\}_{i=1}^{n}$, our goal is to learn a policy $\pi_\theta$ that can generate trajectories following the target distribution $\pi_E$. $\pi_\theta$ can be induced from a state-conditioned flow-based model,

$$\mathrm{d}\boldsymbol{z}_t = v_\theta(\boldsymbol{z}_t, t \mid s)\mathrm{d}t, \ \ \boldsymbol{z}_0 \sim \mathcal{N}(0, I). \tag{1}$$

Here, $s$ is the state and the velocity field is parameterized by a neural network $\theta$ that takes the state as an additional input. To capture the expert distribution with the flow-based model, the velocity field can be trained by minimizing a state-conditioned least-squares objective,

$$L(\theta) = \mathop{\mathbb{E}}_{\substack{(\boldsymbol{s}, \boldsymbol{a}) \sim D \\ \boldsymbol{x}_0 \sim \mathcal{N}(0, I)}} \left[ \int_0^1 \| \boldsymbol{a} - \boldsymbol{x}_0 - v_\theta(\boldsymbol{x}_t, t \mid \boldsymbol{s}) \|_2^2 \, \mathrm{d}t \right], \tag{2}$$

where $\boldsymbol{x}_t$ is the linear interpolation between $\boldsymbol{x}_0$ and $\boldsymbol{x}_1 = \boldsymbol{a}$:

$$\boldsymbol{x}_t = t\boldsymbol{a} + (1 - t)\boldsymbol{x}_0. \tag{3}$$

We should differentiate $\boldsymbol{z}_t$ which is the ODE trajectory in (1) from the linear interpolation $\boldsymbol{x}_t$. They do not overlap unless all trajectories of ODE (1) are straight. See Liu et al. [21] for more discussion.

With infinite data sampled from $\pi_E$, unlimited model capacity and perfect optimization, it is guaranteed that the policy $\pi_\theta$ generated from the learned flow matches the expert policy $\pi_E$ [21].

## 3.2 The Variance-Adaptive Nature of Flow

Typically, to sample from the distribution $\pi_\theta$ at state $s$, we start with a random sample $\boldsymbol{z}_0$ from the Gaussian distribution and simulate the ODE (Eq. (1)) with multi-step ODE solvers to get the action. For example, we can exploit $N$-step Euler discretization,

$$\boldsymbol{z}_{t_{i+1}} = \boldsymbol{z}_{t_i} + \frac{1}{N} v_\theta \left( \boldsymbol{z}_{t_i}, t_i \mid \boldsymbol{s} \right), \ \ t_i = \frac{i}{N}, 0 \le i < N. \tag{4}$$

After running the solver, $\boldsymbol{z}_1$ is the generated action. This solver requires inference with the network $N$ times for decision making in every state. Moreover, a large $N$ is needed to obtain a smaller numerical error.

However, different states may have different levels of difficulty in deciding the potential actions. For instance, when traveling from a city A to another city B, there could be multiple ways for transportation, corresponding to a multi-modal distribution of actions. After the way of transportation is chosen, the subsequent actions to take will be almost deterministic. This renders using a uniform Euler solver with the same number of inference steps $N$ across all the states a sub-optimal solution. Rather, it is preferred that the agent can vary its decision-making process as the state of the agent changes. The challenge is how to quantitatively estimate the *complexity of a state* and employ the estimation to *adjust the inference of the flow-based policy*.

**Variance as a Complexity Indicator** We notice an intriguing property of the policy learned by rectified flow, which connects the complexity of a state with the training loss of the flow-based policy: if the distribution of actions is deterministic at a state $\boldsymbol{s}$ (i.e., a Dirac distribution), the trajectory of rectified flow ODE is a straight line, i.e., *a single Euler step* yields an exact estimation of $\boldsymbol{z}_1$.

**Proposition 3.1.** *Let $v^*$ be the optimum of Eq. (2). If $\mathrm{var}_{\pi_E}(\boldsymbol{a} \mid \boldsymbol{s}) = 0$ where $\boldsymbol{a} \sim \pi_E(\cdot \mid \boldsymbol{s})$, then the learned ODE conditioned on $\boldsymbol{s}$ is*

$$\mathrm{d}\boldsymbol{z}_t = v^*(\boldsymbol{z}_t, t \mid \boldsymbol{s})\mathrm{d}t = (\boldsymbol{a} - \boldsymbol{z}_0)\mathrm{d}t, \ \ \forall t \in [0, 1], \tag{5}$$

*whose trajectories are straight lines pointing to $\boldsymbol{z}_1$ and hence can be calculated exactly with one step of Euler step:*

$$\boldsymbol{z}_1 = \boldsymbol{z}_0 + v^*(\boldsymbol{z}_0, 0 \mid \boldsymbol{s}).$$

**Algorithm 1** AdaFlow: Execution

---

1: **Input:** Current state $s$, minimal step size $\epsilon_{\min}$, error threshold $\eta$, pre-trained networks $v_\theta$ and $\sigma_\phi$.

2: Initialize $\boldsymbol{z}_0 \sim \mathcal{N}(0, I)$, $t = 0$.
3: **while** $t < 1$ **do**
4:     Compute step size
$$\epsilon_t = \text{Clip}\left(\frac{\eta}{\sigma_\phi(\boldsymbol{z}_t, t \mid \boldsymbol{s})}, \quad [\epsilon_{\min}, \ 1 - t]\right).$$
5:     Update   $t \leftarrow t + \epsilon_t, \boldsymbol{z}_t \leftarrow \boldsymbol{z}_t + \epsilon_t v_\theta(\boldsymbol{z}_t, t \mid \boldsymbol{s})$.
6: **end while**
7: Execute action $a = \boldsymbol{z}_1$.

---

Note that the straight trajectories of (5) satisfies $\boldsymbol{z}_t = t\boldsymbol{a} + (1-t)\boldsymbol{z}_0$, which makes it coincides with the linear interpolation $\boldsymbol{x}_t$. As show in [21], this happens only when all the linear trajectories do not intersect on time $t \in [0, 1)$.

More generally, we can expect that the straightness of the ODE trajectories depends on how deterministic the expert policy $\pi_E$ is. Moreover, the straightness can be quantified by a conditional variance metric defined as follows:

$$\sigma^2(x, t \mid \boldsymbol{s}) = \text{var}(\boldsymbol{a} - \boldsymbol{x}_0 \mid \boldsymbol{x}_t = x, \boldsymbol{s}) \tag{6}$$
$$= \mathbb{E}\left[\|\boldsymbol{a} - \boldsymbol{x}_0 - v^*(\boldsymbol{x}_t, t \mid \boldsymbol{s})\|^2 \mid \boldsymbol{x}_t = x, \boldsymbol{s}\right].$$

**Proposition 3.2.** *Under the same condition as Proposition 3.1, we have $\sigma^2(\boldsymbol{z}_t, t \mid \boldsymbol{s}) = 0$ from* (5).

The proof of the above propositions is in Appendix A.1. To summarize, the variance of the state-conditioned loss function at $(\boldsymbol{z}_t, t)$ can be an indicator of the multi-modality of actions. When the variance is zero, the flow-based policy can generate the expected action with only one query of the velocity field, saving a huge amount of computation. In Section 3.3, we will show the variance can be used to bound the discretization error, thereby enabling the design of an adaptive ODE solver.

**Variance Estimation Network**   In practice, the conditional variance $\sigma^2(x, t \mid \boldsymbol{s})$ can be empirically approximated by a neural network $\sigma_\phi^2(x, t \mid \boldsymbol{s})$ with parameter $\phi$. Once the neural velocity $v_\theta$ is learned, we can estimate $\sigma_\phi$ by minimizing the following Gaussian negative log-likelihood loss:

$$\min_\phi \mathbb{E}\left[\int_0^1 \frac{\|\boldsymbol{a} - \boldsymbol{x}_0 - v_\theta(\boldsymbol{x}_t, t|\boldsymbol{s})\|^2}{2\sigma_\phi^2(\boldsymbol{x}_t, t|\boldsymbol{s})} + \log \sigma_\phi^2(\boldsymbol{x}_t, t|\boldsymbol{s})\mathrm{d}t\right]. \tag{7}$$

We adopt a two-stage training strategy by first training the velocity network $v_\theta$ then the variance estimation network $\sigma_\phi$. In practice, the second stage just involves fine-tuning a few linear layers on top of the trained velocity network. Alternatively, we can optimize both the variance estimation and action generation simultaneously, which can extend training time. Our experiments show that joint training and two-stage training yield comparable performance.

### 3.3   Variance-Adaptive Flow-Based Policy

Because the variance indicates the straightness of the ODE trajectory, it allows us to develop an adaptive approach to set the step size to yield better estimation with lower error during inference.

To derive our method, let us consider to advance the ODE with step size $\epsilon_t$ at $\boldsymbol{z}_t$:

$$\boldsymbol{z}_{t+\epsilon_t} = \boldsymbol{z}_t + \epsilon_t v^*(\boldsymbol{z}_t, t \mid \boldsymbol{s}). \tag{8}$$

The problem is how to set the step size $\epsilon_t$ properly. If $\epsilon_t$ is too large, the discretized solution will significantly diverge from the continuous solution; if $\epsilon_t$ is too small, it will take excessively many steps to compute.

We propose an adaptive ODE solver based on the principle of matching the discretized marginal distribution $p_t$ of $\boldsymbol{z}_t$ from (8), and the ideal marginal distribution $p_t^*$ when following the exact ODE

(1). This is made possible with a key insight below showing that the discretization error can be bounded by the conditional variance $\sigma^2(\boldsymbol{z}_t, t \mid s)$.

**Proposition 3.3.** *Let $p_t^*$ be the marginal distribution of the exact ODE $\mathrm{d}\boldsymbol{z}_t = v^*(\boldsymbol{z}_t, t \mid s)\mathrm{d}t$. Assume $\boldsymbol{z}_t \sim p_t = p_t^*$ and $p_{t+\epsilon_t}$ the distribution of $\boldsymbol{z}_{t+\epsilon_t}$ following (8). Then we have*

$$W_2(p_{t+\epsilon_t}^*, p_{t+\epsilon_t})^2 \leq \epsilon_t^2 \mathbb{E}_{\boldsymbol{z}_t \sim p_t}[\sigma^2(\boldsymbol{z}_t, t \mid s)],$$

*where $W_2$ denotes the 2-Wasserstein distance.*

We provide the proof in Appendix A.2. Hence, given a threshold $\eta$, to ensure that an error of $W_2(p_{t+\epsilon_t}^*, p_{t+\epsilon_t})^2 \leq \eta^2$, we can bound the step size by $\epsilon_t \leq \eta/\sigma(\boldsymbol{z}_t, t \mid s)$. Because $\epsilon_t$ at time $t$ should not be large than $1 - t$, we suggest the following rule for setting the step size $\epsilon_t$ at $\boldsymbol{z}_t$ at time $t$:

$$\epsilon_t = \mathrm{Clip}\left(\frac{\eta}{\sigma(\boldsymbol{z}_t, t \mid s)}, \quad [\epsilon_{\min}, \ 1 - t]\right), \tag{9}$$

where we impose an additional lower bound $\epsilon_{\min}$ to avoid $\epsilon_t$ to be unnecessarily small. Besides, the proposed adaptive strategy guarantees to instantly arrive at the terminal point when $\sigma^2(\boldsymbol{z}_t, t \mid s) = 0$ as $\epsilon_t = 1 - t$. Moreover, it aligns with Section. 3.2 since for states with deterministic actions, it sets $\epsilon_0 = 1$ to generate the action in one step. We incorporate the above insights to the execution in Algorithm 1.

**Global Error Analysis** Proposition 3.3 provides the local error at each Euler step. In the following, we provide an analysis of the overall error for generating $\boldsymbol{z}_1$ when we simulate ODE while following the adaptive rule (9). To simplify the notation, we drop the dependency on the state $s$, and write $v_t^*(\cdot) = v^*(\cdot, t \mid \boldsymbol{s})$.

**Assumption 3.4.** *Assume $\|v_t^*\|_{Lip} \leq L$ for $t \in [0, 1]$, and the solutions of $\mathrm{d}\boldsymbol{z}_t = v_t(\boldsymbol{z}_t)\mathrm{d}t$ has bounded second curvature $\|\ddot{\boldsymbol{z}}_t\| \leq M$ for $t \in [0, 1]$.*

This is a standard assumption in numerical analysis, under which Euler's method with a constant step size of $\epsilon_{\min}$ admits a global error of order $O(\epsilon_{\min})$.

**Proposition 3.5.** *Under Assumption 3.4, assume we follow Euler step (8) with step size $\epsilon_t$ in (9), starting from $\boldsymbol{z}_0 = \boldsymbol{x}_0 \sim p_0^*$. Let $p_t$ be the distribution of $\boldsymbol{z}_t$ we obtained in this way, and $p_t^*$ that of $\boldsymbol{x}_t$ in (3). Note that $p_1^*$ is the true data distribution. Set $\eta = M_\eta \epsilon_{\min}^2/2$ for some $M_\eta > 0$, and $\epsilon_{\min} = 1/N_{\max}$.*

*Let $N_{\mathrm{ada}}$ be the number of steps we arrive at $\boldsymbol{z}_1$ following the adaptive schedule. We have*

$$W_2(p_1^*, p_1) \leq C \times \frac{N_{\mathrm{ada}}}{N_{\max}} \times \epsilon_{\min},$$

*where $C$ is a constant depending on $M$, $M_\eta$ and $L$.*

The idea is that the error is proportional to $\frac{N_{\mathrm{ada}}}{N_{\max}}$, suggesting that the algorithm claims an improved error bound in the good case when it takes a smaller number of steps than the standard Euler method with constant step size $\epsilon_{min}$. We provide the proof in Appendix A.3.

**Discussion of AdaFlow and Rectified Flow.** Rectified Flow operates in two stages: the first is learning an ordinary differential equation (ODE), and the second involves a technique called "reflow" used to straighten the learned trajectory. Theoretically, reflow allows for one-step action generation. However, using reflow introduces two major drawbacks: 1) It significantly prolongs training time, particularly because generating the required pseudo noise-data pairs through ODE simulation is computationally expensive; 2) It leads to poorer generation quality due to straightened ODE. In contrast, our method utilizes only the original ODE, eliminating the need for an additional reflow or distillation process, and consistently achieves more accurate action generation.

## 4  Experiments

We conducted comprehensive experiments on four sets of tasks: **1)** a simple 1D toy example to demonstrate the computational adaptivity of AdaFlow; **2)** a 2D navigation problem; and two robot

manipulation task suites on **3)** RoboMimic [14] following past works [6] and **4)** LIBERO [3], provide diverse and realistic scenarios for evaluation.

Our results show that AdaFlow improves the success rate of completing both navigation and manipulation tasks, outperforming state-of-the-art methods such as BC and its variants, as well as Diffusion Policy, across a range of tasks. Additionally, AdaFlow drastically reduces the inference cost. Further experiments demonstrate that AdaFlow is robust to changes in hyperparameters and can adaptively adjust its inference speed according to different states, ensuring efficient and reliable performance.

|  | BC | Diffusion Policy | Rectified Flow | AdaFlow |
|---|---|---|---|---|
| Behavior Diversity | ✗ | ✓ | ✓ | ✓ |
| Fast Action Generation | ✓ | ✗ | ✓ | ✓ |
| No Distillation / Reflow | ✓ | ✓ | ✗ | ✓ |

Table 1: Comparison between BC, Diffusion Policy, Rectified Flow and AdaFlow.

## 4.1 Regression

We start with a 1D regression task designed to demonstrate the adaptivity nature of AdaFlow. The goal is to learn a mapping from $x$ to $y$ where

$$
y = \begin{cases} 0 & \text{for } x \le 0 \\ \pm x & \text{for } x > 0. \end{cases} \tag{10}
$$

Note that $y \mid x$ is deterministic when $x \le 0$ and stochastic otherwise. The training and testing data are uniformly sampled from the ground-truth function with $x \in [-5, 5]$. Details about the setup and the hyperparameters are provided in Appendix.

**AdaFlow can achieve 1-step generation for deterministic states.** Figure 2 (top-right) shows the generation trajectories of Diffusion Policy and AdaFlow with 5 step. Notably, when $x \le 0$, AdaFlow generates *straight* trajectories and is therefore able to predict $y$ with a single step, aligning our analysis in Proposition 3.1 and 3.2. In contrast, Diffusion Policy generates curved trajectories when step = 5, and hence cannot predict $y$ accurately with a single step. The bottom of Figure 2 shows the estimated variance by AdaFlow across $x \in [-5, 5]$, which accurately aligns with the expected variance. In addition, as $x$ increases, AdaFlow adaptively increases the required number of simulation steps.

## 4.2 Navigating a 2D Maze

We create two sets of maze navigation tasks to validate AdaFlow's performance of modeling multi-modal behavior. In particular, we create two *single-task* environments where the agent starts and ends at a fixed point and two *multi-task* environments where the agent can start and end at different points. All four environments are simulated in D4RL Maze2D [54] using MuJoCo. The environments and demonstrations are visualized in Figure 7.

| Method | NFE↓ | Maze 1 | Maze 2 | Maze 3 | Maze 4 |
|---|---|---|---|---|---|
| *Needs reflow* | | | | | |
| Rectified Flow | 1 | 0.82 | 1.00 | 1.00 | 0.80 |
| BC | 1 | **1.00** | **1.00** | 0.92 | 0.76 |
| BC-GMM | 1 | 0.84 | 1.00 | 0.88 | 0.72 |
| Diffussion Policy | 1 | 0.00 | 0.32 | 0.16 | 0.08 |
| Diffussion Policy | 5 | 0.58 | **1.00** | 0.84 | 0.76 |
| Diffussion Policy | 20 | 0.62 | 0.98 | 0.84 | 0.82 |
| AdaFlow | **1.56** | 0.98 | **1.00** | **0.96** | **0.86** |

Table 2: Performance on maze navigation tasks. The table showcases the success rate for each model across different maze complexities. The highest success rate for each task are highlighted in **bold**. NFE denotes Number of Function Evaluations.

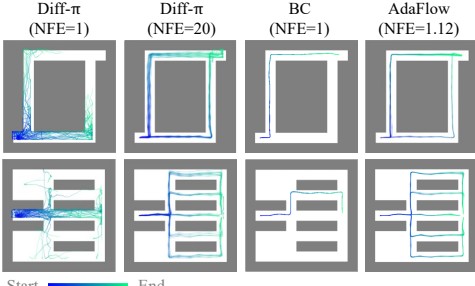

Figure 3: **Generated trajectories.** We visualize the trajectories generated by different policies, with the agent's starting point fixed.

| Method | NFE↓ | Lift | | Can | | Square | | Transport | | ToolHang | Push-T |
|---|---|---|---|---|---|---|---|---|---|---|---|
| | | ph | mh | ph | mh | ph | mh | ph | mh | ph | ph |
| Rectified Flow (*Needs reflow*) | 1 | 1.00 | 1.00 | 0.94 | 1.00 | 0.94 | 0.92 | 0.90 | 0.76 | 0.88 | 0.92 |
| LSTM-GMM | 1 | **1.00** | **1.00** | **1.00** | **1.00** | 0.95 | 0.86 | 0.76 | 0.62 | 0.67 | 0.69 |
| IBC | 1 | 0.79 | 0.15 | 0.00 | 0.01 | 0.00 | 0.00 | 0.00 | 0.00 | 0.00 | 0.75 |
| BET | 1 | **1.00** | **1.00** | **1.00** | **1.00** | 0.76 | 0.68 | 0.38 | 0.21 | 0.58 | - |
| Diffusion Policy | 1 | 0.04 | 0.04 | 0.00 | 0.00 | 0.00 | 0.00 | 0.00 | 0.00 | 0.00 | 0.04 |
| Diffusion Policy | 2 | 0.64 | 0.98 | 0.52 | 0.66 | 0.56 | 0.12 | 0.84 | 0.68 | 0.68 | 0.34 |
| Diffusion Policy | 100 | **1.00** | **1.00** | **1.00** | **1.00** | **1.00** | **0.97** | 0.90 | 0.72 | **0.90** | 0.91 |
| AdaFlow | **1.17** | **1.00** | **1.00** | **1.00** | 0.96 | 0.98 | 0.96 | **0.92** | **0.80** | 0.88 | **0.96** |

Table 3: Success rate on RoboMimic Benchmark. The highest success rate for each task are highlighted in **bold**.

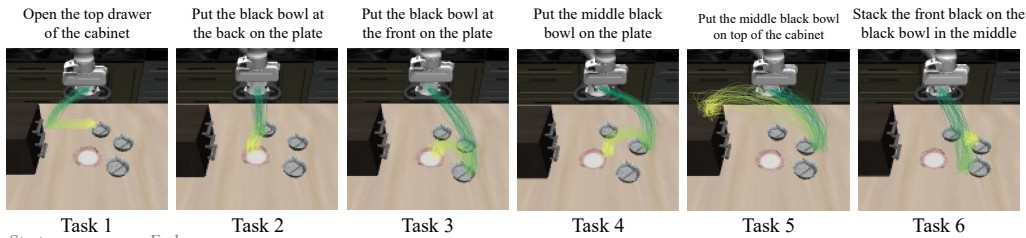

Figure 4: **LIBERO tasks.** We visualize the demonstrated trajectories of the robot's end effector.

**AdaFlow achieves high diversity and success with low NFE; Diffusion Policy and BC lag in comparison.** We compare AdaFlow against baseline methods in Table 2. We additionally visualize the rollout trajectories from each learned policy in Figure 3 as a qualitative comparison of the learned behavior across different methods. From the results, we see that AdaFlow with an average Number of Function Evaluation (NFE) of 1.56 NFE can achieve highly diverse behavior and high success rate in the meantime. By contrast, Diffusion Policy only demonstrates diverse behavior when NFE is larger than 5 and falls behind in success rate even with 20 NFE compared to AdaFlow. BC, on the other hand, has high success rate while performing relatively poorly in terms of behavior diversity.

### 4.3 Robot Manipulation Tasks

**Experiment Setup** To further validate how AdaFlow performs on practical robotics tasks, we compare AdaFlow against baselines on a Push-T task [6], the RoboMimic [10] benchmark (Lift, Can, Square, Transport, ToolHang) and the LIBERO [27] benchmark. For the Push-T task and the tasks in RoboMimic, we follow the exact experimental setup described in Diffusion Policy [6]. Following the Diffusion Policy, we add three additional baseline methods: 1) LSTM-GMM, BC with the LSTM model and a Gaussian mixture head, 2) IBC, the implicit behavioral cloning [12], an energy-based model for generative decision-making, and 3) BET [11]. For the LIBERO tasks, we pick a subset of six Kitchen tasks and follow the setup described in the LIBERO paper (Check Figure 4 for the description of the six tasks).

| Method | NFE↓ | Task 1 | Task 2 | Task 3 | Task 4 | Task 5 | Task 6 | Average |
|---|---|---|---|---|---|---|---|---|
| Rectified Flow (*Needs reflow*) | 1 | 0.90 | 0.82 | 0.98 | 0.82 | 0.82 | 0.96 | 0.88 |
| Diffusion Policy | 1 | 0.00 | 0.00 | 0.00 | 0.00 | 0.00 | 0.00 | 0.00 |
| Diffusion Policy | 2 | 0.00 | 0.58 | 0.36 | 0.66 | 0.36 | 0.32 | 0.38 |
| Diffusion Policy | 20 | 0.94 | **0.84** | **0.98** | 0.78 | 0.82 | 0.92 | 0.88 |
| AdaFlow | **1.27** | **0.98** | 0.80 | **0.98** | **0.82** | **0.90** | **0.96** | **0.91** |

Table 4: Success Rate on LIBERO Benchmark. The highest success rate for each task are highlighted in **bold**.

**AdaFlow consistently outperforms competitors in varied robot manipulation tasks with high efficiency.** The results of the Push-T task and the RoboMimic benchmark are summarized in Table 3. From the table, we observe that AdaFlow consistently achieves comparable or higher success rates across different challenging manipulation tasks, compared against all baselines, with only an average

NFE of 1.17. Note that Diffusion Policy, while showing high success rates using NFE = 100, falls behind when NFE = 1. Results for the six LIBERO tasks are presented in Table 4. Aligning with findings from our previous experiments, AdaFlow once again outperforms BC and Diffusion Policy in terms of success rate with an average NFE of 1.27. We additionally visualize the variance predicted by AdaFlow in Figure 5. It can be seen that the model identifies the high variance when the robot's end-effector is close to the object or target area, matching the variance from the demonstration data.

## 4.4 Ablation Study

We valid how AdaFlow performs against baselines regarding the training and inference efficiency. In addition, we examine how critical the variance estimation network is.

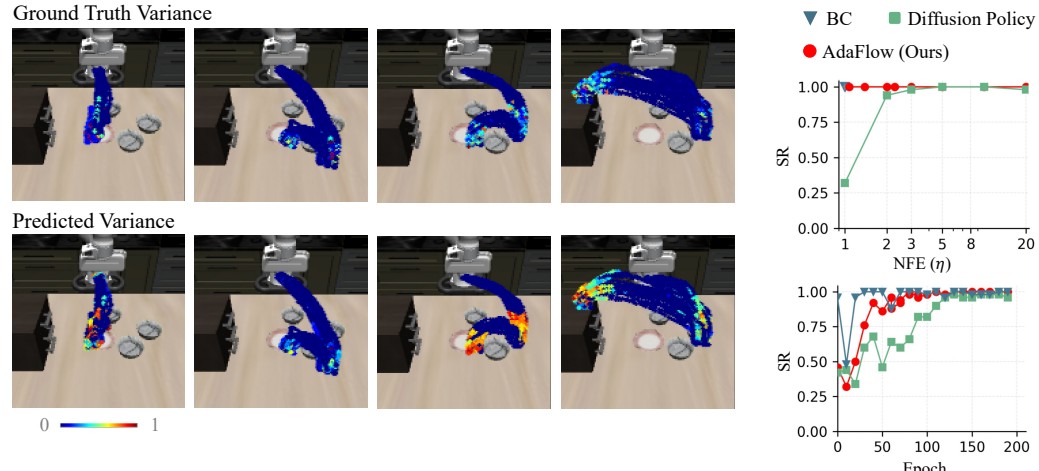

Figure 5: **Predicted variance.** We visualize the variance predicted by AdaFlow. The variance is computed on states from the expert's demonstration and averaged over all simulation steps (e.g., $t$ from 0 to 1). Then we normalize the variance to $[0, 1]$ by the largest variance found at all states.

Figure 6: Ablation studies on AdaFlow.

**Higher Training and Inference Efficiency.** Figure 6 (top) examines changes in success rate relative to the NFE. AdaFlow maintains a high success rate with a very low NFE, whereas the Diffusion Policy generally requires more than three NFE to perform well. Although BC performs well with one NFE, it demonstrates very limited behavioral diversity and struggles to model multi-modal behavior. Figure 6 (bottom) illustrates training efficiency by displaying the success rate over epochs. It shows that AdaFlow has a better area-under-curve than Diffusion Policy, indicating faster learning. As expected, due to its simplicity, Behavioral Cloning (BC) achieves the best learning efficiency.

**Robustness to $\eta$.** In Figure 6, the NFEs in AdaFlow are calculated at various $\eta$ values. It shows that AdaFlow is robust to changes in $\eta$.

**On the Importance of Variance Estimation.** In Table 5, we provide the performance of AdaFlow with and without the variance estimation network on the four mazes from Section 4.2. From the results, it is clear that the variance estimation network not only makes inference faster, but can also lead to better performance.

|  | Maze1 | Maze1 | Maze3 | Maze4 |
|---|---|---|---|---|
| w/o Variance Estimation | 0.78 | 1.00 | 0.92 | 0.80 |
| AdaFlow (Ours) | 0.98 | 1.00 | 0.96 | 0.86 |

Table 5: Ablation study on the use of estimated variance to determine inference steps. Euler sampler is used for AdaFlow without variance estimation.

## 5 Conclusion

We present AdaFlow, a novel imitation learning algorithm adept at efficiently generating diverse and adaptive policies, addressing the trade-off between computational efficiency and behavioral diversity inherent in current models. Through extensive experimentation across various settings, AdaFlow demonstrated superior performance across multiple dimensions including success rate, behavioral diversity, and training/execution efficiency. This work lays a robust foundation for future research on adaptive imitation learning methods in real-world scenarios.

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

# A  Appendix

## A.1  Proof of Proposition 3.1 and Proposition 3.2.

**PROOF 1.** $\text{var}_{\pi_E}(\boldsymbol{a}|s) = 0$ *means that the action* $\boldsymbol{a} = a$ *equals a deterministic value $a$ given s.* *With* $\boldsymbol{x}_t = ta + (1-t)\boldsymbol{x}_0$, *note that*

$$\boldsymbol{a} - \boldsymbol{x}_0 = \frac{1}{1-t}(a - \boldsymbol{x}_t).$$

*Therefore,* $\boldsymbol{a} - \boldsymbol{x}_0$ *is deterministically decided by* $\boldsymbol{x}_t$ *and s. This yields*

$$v^*(x, t \mid s) = \mathbb{E}[a - \boldsymbol{x}_0 \mid \boldsymbol{x}_t = x] = \frac{1}{1-t}(a - x).$$

*Therefore, we have*

$$\mathrm{d}\boldsymbol{z}_t = v^*(\boldsymbol{z}_t, t \mid s) = \frac{1}{1-t}(a - \boldsymbol{z}_t)\mathrm{d}t.$$

*Solving ODE this yields*

$$\boldsymbol{z}_t = ta + (1-t)\boldsymbol{z}_0 = (1-t)v^*(\boldsymbol{z}_0, 0 \mid s).$$

*Differentiating it also yields*

$$\boldsymbol{z}_t = (a - \boldsymbol{z}_0)\mathrm{d}t.$$

*We also have* $\sigma^2(x, t \mid s)$ *again because* $a - \boldsymbol{x}_0$ *is deterministic given* $\boldsymbol{x}_t$ *and s:*

$$\sigma^2(x, t \mid s) = \text{var}(\boldsymbol{a} - \boldsymbol{x}_0 \mid \boldsymbol{x}_t = x, s) = 0.$$

## A.2  Proof of Proposition 3.3

**PROOF 2.** *Following the property of rectified flow, the distribution of* $\boldsymbol{x}_1 = ta + (1-t)\boldsymbol{x}_0$ *coincides with* $p_t$ *for all* $t \in [0, 1]$. *Hence, we can assume that* $\boldsymbol{z}_t = \boldsymbol{x}_t \sim p_t^*$. *In this case, we have* $\boldsymbol{z}_{t+\epsilon_t} = \boldsymbol{x}_t + \epsilon_t v^*(\boldsymbol{z}_t, t \mid s)$ *and* $\boldsymbol{x}_{t+\epsilon_t} = \boldsymbol{x}_t + \epsilon_t(\boldsymbol{a} - \boldsymbol{x}_0)$. *We have*

$$W_2(p_{t+\epsilon_t}^*, p_{t+\epsilon_t})^2$$

$$\leq \mathbb{E}\left[\|\boldsymbol{z}_{t+\epsilon_t} - \boldsymbol{x}_{t+\epsilon_t}\|_2^2\right]$$

$$= \mathbb{E}\left[\mathbb{E}\left[\|\boldsymbol{z}_{t+\epsilon_t} - \boldsymbol{x}_{t+\epsilon_t}\|_2^2 \mid \boldsymbol{x}_t\right]\right]$$

$$= \mathbb{E}\left[\mathbb{E}\left[\|\epsilon_t v^*(\boldsymbol{z}_t, t \mid s) - \epsilon_t(\boldsymbol{a} - \boldsymbol{x}_0)\|_2^2 \mid \boldsymbol{x}_t\right]\right]$$

$$= \epsilon_t^2 \mathbb{E}_{\boldsymbol{z}_t \sim p_t}[\sigma^2(\boldsymbol{z}_t, t \mid s)].$$

## A.3  Proof of Proposition 3.5

**PROOF 3.** *Assume the adaptive algorithm visits the time grid of* $0 = t_0, t_1, \ldots, t_N = 1$.

*Define* $\boldsymbol{z}_t^{t_i}$ *be the result when we implement the adaptive discretization algorithm upto* $t_i$ *and then switch to follow the exact ODE afterward, that is, we have* $\mathrm{d}\boldsymbol{z}_t^{t_i} = v_t(\boldsymbol{z}_t^{t_i})\mathrm{d}t$ *for* $t \geq t_i$. *In this way, we have* $\boldsymbol{z}_t^1 = \boldsymbol{z}_t$, *and* $\boldsymbol{z}_t^0 = \boldsymbol{z}_t^*$, *where* $\boldsymbol{z}_t^*$ *is the trajectory of the exact ODE* $\mathrm{d}\boldsymbol{z}_t^* = v_t^*(\boldsymbol{z}_t^*)\mathrm{d}t$.

*From Lemma A.1, we have*

$$\left\|\boldsymbol{z}_1^{t_{i-1}} - \boldsymbol{z}_1^{t_i}\right\| \leq \exp(L(1 - t_i))\left\|\boldsymbol{z}_{t_i}^{t_i} - \boldsymbol{z}_{t_i}^{t_{t-1}}\right\|.$$

*Let* $p_t^{t_i}$ *be the distribution of* $\boldsymbol{z}_t^{t_i}$. *Then we have* $p_t^1 = p_t$ *and* $p_t^0 = p_t^*$. *Then*

$$W_2(p_1^{t_{i-1}}, p_1^{t_i}) \leq \mathbb{E}\left[\left\|\boldsymbol{z}_1^{t_{i-1}} - \boldsymbol{z}_1^{t_i}\right\|^2\right]^{1/2}$$

$$= \exp(L(1 - t_i))\mathbb{E}\left[\left\|\boldsymbol{z}_{t_i}^{t_{i-1}} - \boldsymbol{z}_{t_i}^{t_i}\right\|^2\right]^{1/2}$$

$$= \exp(L(1 - t_i))\max(\eta, \ \epsilon_{min}^2 M/2)$$

$$= C\epsilon_{\min}^2 \exp(-Lt_i),$$

where $C = \frac{1}{2}\max(M, M_\eta)\exp(L(1 - t_i))$. *Here we use the bound in the proof of Proposition 3.1 and Lemma A.1. Hence,*

$$W_2(p_1^*, p_1) = \sum_{i=1}^{N_{\text{ada}}} W_2(p_1^{t_{i-1}}, p_1^{t_i})$$

$$\leq \sum_{i=1}^{N_{\text{ada}}} C\epsilon_{\min}^2 \exp(-Lt_i)$$

$$\leq C \times \frac{N_{\text{ada}}}{N_{\max}} \times \epsilon_{\min},$$

*where* $C = \exp(L)\max(M, M_\eta)$.

**Lemma A.1.** *Let* $\|v_t\|_{Lip} \leq L$ *for* $t \in [0, 1]$. *Assume* $x_t$ *and* $y_t$ *solve* $\mathrm{d}x_t = v_t(x_t)\mathrm{d}t$ *and* $\mathrm{d}y_t = v_t(y_t)\mathrm{d}t$ *starting from* $x_0, y_0$, *respectively. We have*

$$\|x_t - y_t\| \leq \exp(Lt)\|x_0 - y_0\|, \quad \forall t \in [0, 1]. \tag{11}$$

**PROOF 4.**

$$\frac{\mathrm{d}}{\mathrm{d}t}\|x_t - y_t\|^2 = 2(x_t - y_t)^\top(v_t(x_t) - v_t(y_t))$$

$$\leq 2L\|x_t - y_t\|^2,$$

*where we used* $\|v_t(x_t) - v_t(y_t)\| \leq L\|x_t - y_t\|$. *Using Gronwall's inequality yields the result.*

**Lemma A.2.** *Under Assumption 3.4, we have*

$$\|x_{t+\epsilon} - (x_t + \epsilon v_t(x_t))\| \leq \frac{\epsilon^2 M}{2},$$

*for* $0 \leq t \leq \epsilon + t \leq 1$.

**PROOF 5.** *Direct application of Taylor approximation.*

## A.4 Visualization of Tasks

We provide a visualization of the 2D Maze Figure 7.

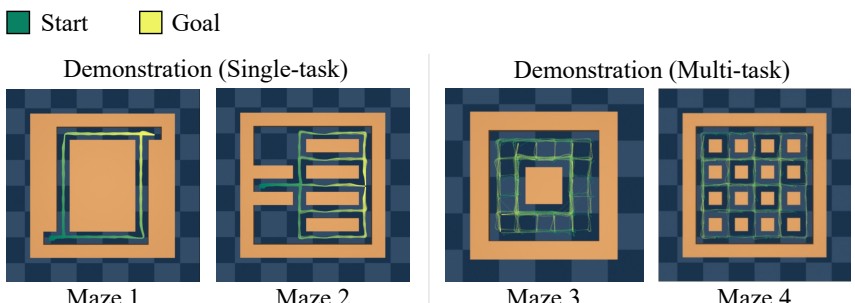

Figure 7: Trajectories of 100 demonstrations for each maze.

## A.5 Planner for Maze2D task

We generate the demonstration data in Maze toy using planner similar to [54]. The planner devises a path in a maze environment by calculating waypoints between the start and target points. It begins by transforming the given continuous-state space into a discretized grid representation. Employing Q-learning, it evaluates the optimal actions and subsequently computes the waypoints by performing a rollout in the grid, introducing random perturbations to the waypoints for diversity. The controller connects these waypoints in an ordered manner to form a feasible path. In runtime, it dynamically adjusts the control action based on the proximity to the next waypoint and switches waypoints when close enough, ensuring the trajectory remains adaptive and efficient.

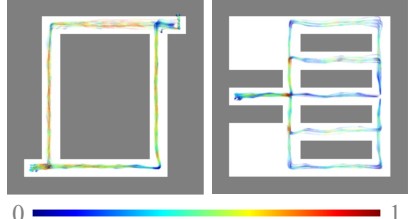

Figure 8: Predicted variance by AdaFlow on the Maze task.

## A.6 Comparative Analysis of Separate and Joint Training

In this section, we provide a comparison between the two training strategies employed in our proposed solution: separate training and joint training. Our primary objective is to investigate whether there is a substantial difference in performance and efficiency between these two training approaches.

**Experiment Setup.** To conduct this comparative analysis, we designed experiments using our proposed framework with both training strategies. Specifically, we consider two approaches: separate and joint training. In **Separate Training** setting, we train the variance prediction network and the policy function separately, as described in our main paper. In **Joint Training** setting, we train both the variance prediction network and the policy function simultaneously in an end-to-end manner. The goal is to assess the impact of these training strategies on the overall performance.

**Results and Discussion.** As shown in Table 6, the performance were consistent between the two training approaches, indicating the effectiveness of our two-stage framework in balancing policy accuracy and uncertainty estimation. Separate training exhibited faster computational speed, making it the preferred choice once the policy function was robustly trained. Joint training required more computational resources and time.

|                     | Maze 1 | Maze 2 | Maze 3 | Maze 4 |
|---------------------|--------|--------|--------|--------|
| AdaFlow (Separate)  | 0.98   | 1.00   | 0.96   | 0.86   |
| AdaFlow (Joint)     | 1.00   | 1.00   | 0.96   | 0.88   |

Table 6: Performance comparison of separate training and joint training of AdaFlow in Maze tasks.

## A.7 Visualization of Exact Variance.

In the main paper, we showed the variance predictions made by AdaFlow across different states within a robot's state space. Here, we explain how we compute the *exact* variance for different states, to provide a ground truth of variance for reference. To achieve this, we first train a 1-Rectified Flow model for the task, then we can compute the exact variance by sampling:

$$\frac{1}{N_t}\frac{1}{N_z}\sum_t\sum_{\boldsymbol{z}_0}\mathbb{E}\big[||y-\boldsymbol{z}_0-v(\boldsymbol{z}_t,t;x)||^2\big], \ \ \text{where} \ \ \boldsymbol{z}_t=ty+(1-t)\boldsymbol{z}_0,(x,y)\sim p^*. \quad (12)$$

For each states, we randomly sample 10 time steps ($N_t = 10$) and 10 noises ($N_z = 10$).

## A.8 Visualization of Predicted Variance on Maze task.

We present the predicted variance by AdaFlow in Figure 8.

## A.9 Additional Experimental Details.

**Model Architectures.** For the 1D toy example, we used a MLP constructed with 5 fully connected layers and SiLU activation functions. We integrated temporal information by extending the time input into a 100-dimensional time-encoding vector through the cosine transformation of a random vector, $cost * z_T$, where $z_T$ is sampled from a Gaussian distribution. This time feature is then concatenated

with the noise and condition ($x$) inputs to for time-aware predictions. The network comprises 4 hidden layers, each with 100 neurons, and the output layer predict a single $y$ value. The dataset consists of 10000 single-dimensional samples uniformly distributed in the range $[-5, 5]$.

For navigation and robot manipulation tasks, we adopted the model architecture from Diffusion Policy [6]. For navigation task, we use the same architecture as used in Push-T task. In the RoboMimic and LIBERO experiments, we used the Diffusion Policy-C architecture. To ensure a fair comparison across different methods, we maintained a consistent architecture for all methods in our experiments, except where specifically noted. Detailed parameters are available in Table 7.

| Hyperparameter | 1D Toy | | | Maze | | | RoboMimic & LIBERO | | |
|---|---|---|---|---|---|---|---|---|---|
| | RF & AdaFlow | BC | Diffusion Policy | RF & AdaFlow | BC | Diffusion Policy | RF & AdaFlow | BC | Diffusion Policy |
| Learning rate | 1e-2 | 1e-2 | 1e-2 | 1e-4 | 1e-4 | 1e-4 | 1e-4 | 1e-4 | 1e-4 |
| Optimizer | Adam | Adam | Adam | AdamW | AdamW | AdamW | AdamW | AdamW | AdamW |
| $\beta_1$ | 0.9 | 0.9 | 0.9 | 0.9 | 0.9 | 0.9 | 0.95 | 0.95 | 0.95 |
| $\beta_2$ | 0.999 | 0.999 | 0.999 | 0.999 | 0.999 | 0.999 | 0.999 | 0.999 | 0.999 |
| Weight decay | 0 | 0 | 0 | 1e-6 | 1e-6 | 1e-6 | 1e-6 | 1e-6 | 1e-6 |
| Batch size | 1000 | 1000 | 1000 | 256 | 256 | 256 | 64 | 64 | 64 |
| Epochs | 200 | 200 | 400 | 200 | 200 | 200 | 500(L) / 3000(RM) | 500(L) / 3000(RM) | 500(L) / 3000(RM) |
| Learning rate scheduler | cosine | cosine | cosine | cosine | cosine | cosine | cosine | cosine | cosine |
| EMA decay rate | - | - | - | 0.9999 | 0.9999 | 0.9999 | 0.9999 | 0.9999 | 0.9999 |
| Number of training time steps | - | - | 100 | - | - | 20 | - | - | 100 |
| Number of Inference time steps | 100 (RF) | - | 100(DDPM) | - | - | 20(DDPM) | - | - | 100(DDPM) |
| $\eta$ | 0.1 | - | - | 1.5 | - | - | 1.0 | - | - |
| $\epsilon_{min}$ | 5 | - | - | 5 | - | - | 10 | - | - |
| Action prediction horizon | - | - | - | 16 | 16 | 16 | 16 | 16 | 16 |
| Number of observation input | - | - | - | 2 | 2 | 2 | 2 | 2 | 2 |
| Action execution horizon | - | - | - | 8 | 8 | 8 | 8 | 8 | 8 |
| Observation input size | 1 | 1 | 1 | 4 (Single-task) / 6(Multi-task) | | | $76 \times 76$ | $76 \times 76$ | $76 \times 76$ |

Table 7: Hyperparameters used for training AdaFlow and baseline models.

**Implementation of Baselines.** In our studies, **BC** was implemented as a baseline, applying behavior cloning in its most straightforward form and using a Mean Squared Error loss function between the predicted and ground truth actions. The implementations for DDPM and DDIM remained consistent with those outlined in [6]. Across all experiments, consistency was maintained regarding architecture, input, and output, with all methods adhering to a similar experimental pipeline. We just use a 4 layer MLP with SiLU activation for the variance prediction, with hidden dimension of 512, which is a very small network whose computational overhead can be neglected compared to the full model.

**Implementation of Vairance Prediction Network.** In the 1D toy experiment, we designed the variance prediction network as a 4-layer MLP, mirroring the main model's architecture for simplicity. In theory, the variance estimation network takes the same input as rectified flow model, so its input can be just the intermediate features extracted by the main model. Hence in the navigation and manipulation experiments, the inputs of variance prediction networks are the bottle-neck features extracted by the U-Net model.

**Training on RoboMimic.** Training Diffusion Models on RoboMimic is very resource-intensive. Training and evaluating a Transport task requires over a month of GPU hours. More complex tasks, such as ToolHang, can demand up to three times longer [1]Given the challenges in replicating the results from [6], we opted to start with their open-sourced pretrained model. We then fine-tuned the baselines and our method for 500 epochs and subsequently compared the performance of different models.

## A.10 Comparison with standard Rectified Flow.

For the purpose of policy learning, we can consider standard Rectified Flow as a subset of our method, which can be recovered with specific choices of $\eta$ and $\epsilon_{min}$. In this section, we compare our approach with the standard Rectified Flow, particularly focusing on the generation within a single step. Standard Rectified Flow requires a reflow or distillation stage to straighten the ODE process. During this reflow stage, the model simulates data using the initial 1-Rectified Flow. These data are then used in distillation training, resulting in what is termed a 2-Rectified Flow. Theoretically, a 2-Rectified Flow is capable of producing a straight generation trajectory, which enables one-step generation. In contrast, the 1-Rectified Flow tends to be less straight, necessitating multiple steps for sample generation.

In Table 8, we compare the performance of 1-Rectified Flow, 2-Rectified Flow, and our method in the maze task. Furthermore, Figure 9 illustrates the trajectories produced by both standard Rectified Flow

---

[1]See this link

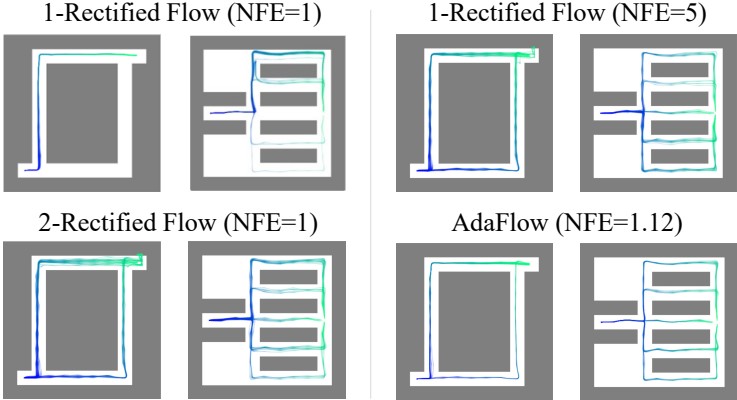

Figure 9: **Generated trajectories.** We visualize the trajectories generated by standard Rectified Flow and AdaFlow, with the agent's starting point remaining fixed. 0

and our method. It's evident that the standard 1-Rectified Flow struggles to generate a diverse range of actions in a single step. In contrast, our method is able to produce diverse behaviors in nearly one step. This efficiency is attributed to our method's ability to estimate the variance across different states, identifying those that require multi-step generation.

|  | NFE↓ | Maze 1 | Maze 2 | Maze 3 | Maze 4 |
|---|---|---|---|---|---|
| 1-RF | 1 | **1.00** | **1.00** | 0.98 | 0.80 |
| 1-RF | 5 | 0.82 | **1.00** | 0.94 | 0.80 |
| 2-RF (reflow) | 1 | 0.82 | **1.00** | **1.00** | 0.80 |
| AdaFlow ($\eta = 1.5$) | 1.56 | 0.98 | **1.00** | 0.96 | **0.86** |
| AdaFlow ($\eta = 2.5$) | 1.12 | **1.00** | **1.00** | 0.94 | 0.78 |

Table 8: Performance on maze navigation tasks. The table showcases the success rate (**SR**) for each model across different maze complexities. The highest success rate for each task are highlighted in **bold**. Note that 2-RF needs an expensive distillation training stage.

