# OpenReview forum: "AdaFlow: Imitation Learning with Variance-Adaptive Flow-Based Policies"
_NeurIPS.cc/2024/Conference — NeurIPS 2024 poster_

### Official Review · Reviewer_RJmb · 2024-06-27

**Soundness:** 3
**Presentation:** 2
**Contribution:** 3
**Rating:** 6
**Confidence:** 3

**Summary:**

This paper focuses on accelerating flow-based imitation learning algorithms by extending the existing work of Rectified Flow to the field of imitation learning, proposing AdaFlow. The paper leverages probability flows to train a state-conditioned action generator and introduces a variance-adaptive ODE solver to adjust the step size, accelerating inference while ensuring diversity. Experimental results in four environments, including navigation and manipulation, demonstrate the efficiency of AdaFlow.

**Strengths:**

- The idea of this paper is well-motivated and the investigated problem is practical.
- The theme of this paper in speeding up flow-based generative models is timely and interesting.
- Theoretical analysis is complete.
- Empirical results show improvement over baseline and other compared methods. The visualized results are good to demonstrate the efficiency.

**Weaknesses:**

- There are some minor typos in the paper.
- The implementation codes are not provided.
- The experimental data lack statistical significance.
- The limitations of the method are not discussed.

**Questions:**

- Does the step size depend not only on the variance (number of potential behaviors corresponding to the same state) but also on the complexity of expert behaviors?
- In Tables 2, 3, and 4, Reflow achieves comparable (sometimes better) success rates to AdaFlow with NFE=1, which raises doubts about the advantages of AdaFlow. Additionally, can Reflow+distillation achieve even better results as described in [1]?
- The top and bottom in Fig. 6 do not match the description in line 269. Furthermore, as shown in Fig. 6, Diffusion Policy achieves SR=1 at NFE=3~5, contradicting the claim of "requiring 10x less inference times to generate actions."
- The experiments only use success rate as an evaluation metric; however, commonly used metrics in clustering algorithms, such as NMI and ENT, can be considered as measures of behavioral richness.
- What does "Hz" mean in Table 1?
- In formula (7), the log likelihood should be $\log(\sigma^2)$ instead of $\log(\sigma)$.
- Please carefully check and correct the formatting of references, e.g., [36] lacks author information, and references like [15], [32], [34] lack dates.

[1] Liu X, Gong C, Liu Q. Flow straight and fast: Learning to generate and transfer data with rectified flow[J]. arXiv preprint arXiv:2209.03003, 2022.

**Limitations:**

The paper lacks a discussion on the limitations of the method.

---

> ### Author Rebuttal · Authors · 2024-08-07
>
> Thank you for your valuable feedback, which has helped us improve the clarity and comprehensiveness of our work.
>
> **Complexity of expert behavior and variance**
>
> The complexity of expert behavior and the variance are intertwined. Since we perform offline learning with demonstration data, the model learns a high variance when the expert’s behavior becomes diverse at a given state, and vice versa. So even if the expert behavior is very complicated (like peg in the hole), but the action itself is relatively deterministic (there is a consistent way of solving the underlying task), then AdaFlow will still choose a large step size, because the model is more certain about which action to take, so it performs more like in the BC case. In contrast, even if it is a simple task like lifting an object, if there exists many different ways of lifting, then AdaFlow will still choose a small step size, until the lifting behavior is determined (the conditional variance drops accordingly).
>
> **Comparison with Reflow**
>
> Yes, Reflow can achieve comparable success rates to AdaFlow. However, Reflow requires an additional complex and time-consuming stage after training the original flow. AdaFlow is much more efficient than Reflow. This is because:
>
> - AdaFlow doesn’t have to be trained in two separate stages, **it can also learn action generation and variance estimation together**. As shown in the following table (originally in Appendix A.6), training the flow and the variance networks separately or jointly results in similar performance. In practice, we chose to train them separately because separate training in fact results in even better training efficiency (faster convergence), this is because the objective of the variance network is stationary. In addition, training the variance network is way faster than training the original flow, hence Adaflow introduces almost no overhead to the 1-Rectified Flow (without reflow). **Compared to AdaFlow, reflow and consistency distillation always require a stage to distill the ODE，which itself alone is more expensive than the original training phase.**
>
> 	|  | Maze 1 | Maze 2 | Maze 3 | Maze 4 |
> 	| - | ------ | ------ | ------ | ------ |
> 	| AdaFlow (Separate) | 0.98 | 1.00 | 0.96 | 0.86 |
> 	| AdaFlow (Joint)    | 1.00 | 1.00 | 0.96 | 0.88 |
>
> - Reflow is more complicated and computationally expensive than AdaFlow. Reflow operates in two steps: first, it generates pseudo data, requiring simulating the full trajectory, i.e., 100 model inferences per pseudo sample, and the reflow dataset usually needs to be as large as the original dataset. Second, it trains the new flow model with a reflow target, which might introduce errors. AdaFlow doesn’t need this expensive data generation. If we train the variance estimation part separately (it can also be trained jointly), it only involves training a few linear layers on top of the original rectified flow model. In practice, we compared the total training time for the reflow stage and AdaFlow’s variance estimation training stage. On the maze task, we observed that reflow takes **7.5x longer than AdaFlow in terms of wall clock time.**
>
> - It is possible to apply a distillation stage after reflow. In image and video generation tasks, distillation is often necessary because the ODE might not be straight even after reflow. However, in our robotics tasks, reflow has already learned a very straight trajectory. We observed no further improvements when using more than one inference step for reflow. Therefore, the potential improvement from distillation is very minimal. Additionally, distillation requires another training stage, which adds complexity and inconvenience.
>
> **Measure behavior richness**
>
> We appreciate the suggestion to use NMI or ENT as a measure of behavioral richness. However, it's not feasible to apply them directly to the generated trajectories because each trajectory has a different length. Clustering methods like NMI typically require fixed-length data, which isn't the case with our generated trajectories. We are open to exploring alternative metrics that could better capture the diversity and richness of behaviors in future work. If you have any suggestions or additional metrics that you believe could be useful, we would be happy to consider them.
>
> **Typos**
>
> There’s a typo in Table 1, which we will correct in the revised version. We will correct other typos mentioned by the reviewer in the paper and will reflect these changes in the revised version.
>
> Please let us know if you have any further questions or concerns. We hope that our rebuttal addresses all your points thoroughly, and we would greatly appreciate your reconsideration of the score. Thank you!

---

> > ### Comment · Reviewer_RJmb · 2024-08-10
> > **Thanks for your clarification**
> >
> > Thanks for your rebuttal. My concerns have been resolved, except for the third point in Weaknesses. The standard deviation or error bars of the experimental results need to be clearly provided to avoid the randomness. Anyway, I think this is an interesting job for the field of imitation learning. I decided to improve my score.

---

### Official Review · Reviewer_AbSH · 2024-07-11

**Soundness:** 3
**Presentation:** 3
**Contribution:** 3
**Rating:** 6
**Confidence:** 3

**Summary:**

This paper introduces AdaFlow, a Flow-based policy for imitation learning. The key innovation is a variance-adaptive solver that employs smaller step sizes when complexity is high and reduces to one-step generation when straightforward. The authors provide theoretical analysis, offering an error bound based on a threshold hyperparameter η. Experiments on maze navigation and robot manipulation tasks demonstrate AdaFlow's effectiveness, generating higher success rates and more diverse trajectories compared to popular Diffusion Policies, while requiring significantly fewer NFEs

**Strengths:**

- The paper proposes a simple yet efficient adaptive scheme to dramatically reduce the computational cost for inference in generative model policies.
- The method is well-grounded in theoretical analysis.
- The paper is generally well-written and clearly structured.

**Weaknesses:**

- The environments tested in this work are standard but relatively low-dimensional. It's unclear how well the proposed method scales to high-dimensional tasks like humanoid control.
- The experimental section could benefit from more detailed information about the setup and evaluation metrics.

**Questions:**

- In Table 2 and Table 8, you mention a "diversity score," but I cannot find it in the tables. Could you clarify this discrepancy and explain how you quantify the diversity of generated trajectories?
- How many seeds did you use to train the policies, and how many runs did you use to evaluate each policy? This information seems to be missing from the text.
- How does AdaFlow compare with classical adaptive ODE solvers?
- While AdaFlow doesn't need to perform reflow as in Rectified Flow, it requires training an extra variance estimation network. Could you provide a comparison of the computational costs for training these two methods?
- What is the robustness of AdaFlow to potential errors in the variance estimation network?

**Limitations:**

No, the author doesn't have a discussion about the limitation. As I mentioned in the weakness section, I think the method may have some problems to scale to higher-dimensional action space where learning a ODE and the variance estimation won't be so easy.

---

> ### Author Rebuttal · Authors · 2024-08-07
>
> Thank you for your valuable feedback, which has helped us improve the clarity and comprehensiveness of our work.
>
> **Possibility of extending to higher-dimensional robotics tasks**
>
> We agree that applying AdaFlow to more complex and high-dimensional tasks like humanoid control would be interesting. Due to the time constraints of the rebuttal period, we couldn't include such experiments. However, on the LIBERO dataset, our input is video data plus the gripper configuration of the robot hand, which is one of the most high-dimensional tasks in robot manipulation. In future work, we plan to explore using AdaFlow on humanoid tasks. We are confident that training an ODE on higher-dimensional spaces is feasible, as ODE training has been successful in image generation tasks (as in Stable Diffusion 3)
>
> **Detailed experiment setup**
>
> We will update the paper to include more detailed information about the experimental setup and evaluation metrics in Appendix A.9 Additional Experimental Details. Additionally, we will provide the codebase to assist with reproducing the results.
>
> **Correction of typos**
>
> We have corrected the typos in the paper to ensure consistency and will reflect these changes in the revised version.
>
> **Seeds and runs**
>
> Due to the limited time during the rebuttal period, we couldn't complete the experiments with multiple seeds and runs. However, the partial results we obtain indicate that the performance of AdaFlow, diffusion policy, and other baselines are consistent with what we presented in the paper. We will include the results with multiple seeds in the revision to provide a comprehensive evaluation.
>
> **Comparison with classical adaptive ODE solvers**
>
> We conducted experiments to compare AdaFlow with recent advanced and faster solvers like DPM-solver and DPM-solver++. As shown in the table below, we compared AdaFlow with DPM and DPM++ across multiple tasks on LIBERO. The results demonstrate that AdaFlow achieves higher success rates even with fewer NFE:
>
> |   | NFE | Task 1 | Task 2 | Task 3 | Task 4 | Task 5 | Task 6 | Average |
> | - | ---   | ------ | ------ | ------ | ------ | ------ | ------ | ------- |
> | DPM     | 1    | 0.00 | 0.00  | 0.00   | 0.00   | 0.00   | 0.00  |  0.00 |
> | DPM     | 2    | 0.42 | 0.44  | 0.54   | 0.36   | 0.58   | 0.38  |  0.45 |
> | DPM     | 20   | 1.00 | 0.80  | 0.98   | 0.84   | 1.00   | 0.88  |  0.92 |
> | DPM++   | 1    | 0.00 | 0.00  | 0.00   | 0.00   | 0.00   | 0.00  |  0.00 |
> | DPM++   | 2    | 0.42 | 0.44  | 0.54   | 0.36   | 0.58   | 0.36  |  0.45 |
> | DPM++   | 20   | 1.00 | 0.80  | 0.98   | 0.84   | 1.00   | 0.92  |  0.92 |
> | AdaFlow | 1.27 | 0.98 | 0.80  | 0.98   | 0.82   | 0.90   | 0.96  |  0.91 |
>
>
> **Robustness to potential errors in the variance estimation network**
>
> We acknowledge that the variance estimation network may have errors. As shown in [this figure](https://anonymous.4open.science/r/adaflow_rebuttal-C0D6/variance_error.png), the predicted variance does not perfectly match the ground truth state variance. This estimation error can potentially influence the generation results.
>
> However, note that in Algorithm 1, $\epsilon_t$ is clipped between $\epsilon_\text{min}$ and $(1 - t)$, so in theory, AdaFlow’s worst case performance (with arbitrarily bad variance estimation) will be between 1-step RF and $1 / \epsilon_\text{min}$-step RF.
>
> Empirically, we compare AdaFlow against RF with 20 inference steps on the LIBERO task suites, the results are shown in the following:
>
> |   | NFE | Task 1 | Task 2 | Task 3 | Task 4 | Task 5 | Task 6 | Average |
> | - | --- | ------ | ------ | ------ | ------ | ------ | ------ | ------- |
> | Rectified Flow | 20   | 0.96 | 0.86  | 1.00   | 0.82   | 0.86   | 0.98  |  0.92 |
> | AdaFlow        | 1.27 | 0.98 | 0.80  | 0.98   | 0.82   | 0.90   | 0.96  |  0.91 |
>
> As shown, AdaFlow's performance is comparable to Rectified Flow with NFE=20. This demonstrates that the introduction of the variance estimation process does not significantly impact performance. AdaFlow remains effective and efficient.

---

> > ### Comment · Reviewer_AbSH · 2024-08-10
> > **Thanks for the additional experiments**
> >
> > Thanks for the rebuttal. Some of my concerns have been solved.
> >
> > I appreciate the additional comparison with DPM solvers.  Regarding the higher-dimensional tasks, I don't think the LIBERO dataset is a convincing example. Correct me if I am wrong, in LIBERO dataset, although images are part of the inputs, they are only served as a condition. The output is still the low-dimensional action space. Also, given your reply to Reviewer df3a about AdaFlow on image generation, it seems like AdaFlow will offer a marginal improvement on high-dimensional tasks.
> >
> > Regarding the clip of $\epsilon_t$, it cannot prevent the case that the true variance is high, but the estimation is low.
> >
> > In the light of this, I would keep my score.

---

### Official Review · Reviewer_FRk8 · 2024-07-11

**Soundness:** 2
**Presentation:** 2
**Contribution:** 2
**Rating:** 6
**Confidence:** 3

**Summary:**

In this paper, the authors propose a new step schedule for sampling from a gradient flow that they claim accelerates the sampling process.    While they focus on the setting of Imitation Learning, in principle, their algorithm samples from some conditional distribution $\pi_E$ of actions given observations on Euclidean space by first sampling $z_0$ a standard Gaussian and then evolving $z_t$ according to a gradient flow with velocity field $v_\theta$ learned from data.  As the flow itself has to be discretized, there is a natural question as to how the discrete steps should be chosen.  Whereas many prior works rely on an Euler discretization with uniform step sizes, the authors propose using the conditional variance of the deviation of the action from the endpoint of a linearized flow as a measure of how large a step to take.  Under extremely strong assumptions, the authors demonstrate that their step size leads to control under squared Wasserstein distance between samples from their discretized flow and the desired sample.  They then empirically demonstrate the efficacy of their techniques in a variety of environments.

**Strengths:**

The paper is concerned about the inference speed of learned imitator policies, which is an important problem that is sometimes ignored in academic work on IL, especially with the recent focus on diffusion models.  The authors present a diverse set of examples that is somewhat suggestive of the efficacy of their techniques.

**Weaknesses:**

The "theoretical analysis" (Line 59) presented is imprecise and holds only under extremely strong and unrealistic conditions.  As an example of the imprecision, the conditioning in the definition of the conditional variance in line 147 is a bit confusing to me.  Is the conditioning on $x_t$ intended to be conditional on the path up to time $t$?  If not, then it is not obvious to me that we could expect that $\sigma^2$ can be zero in any setting where the gradient flow converges, due to the variance coming from $x_0$.  If so, this should be formally defined.  Furthermore, many of the claimed propositions, such as 3.1 and 3.2 are immediate.  Proposition 3.3 has a non-rigorous proof that refers to a vague property of a rectified flow without stating precisely what this property is or citing the result.  While in principle, any estimator of the conditional variance would be acceptable, the authors only present one that assumes the conditional distributions of the actions are Gaussian with no justification and no discussion of this assumption.

**Questions:**

See weaknesses.

**Limitations:**

The authors do not address the limitations of their approach, despite the fact that estimating the conditional variance is a potentially challenging problem and their Gaussianity assumption in their suggested approach is potentially limiting.

---

> ### Author Rebuttal · Authors · 2024-08-07
>
> We believe the reviewer might have some misunderstandings, and we will ensure to further clarify our statements in the updated version of the paper.
>
> - **In Line 147**, $x_t$ is a random variable that follows the marginalized probability at time $t$. Thus, $x_t$ is not conditional on the path up to time $t$ (similar to the original Rectified Flow paper). The conditional variance $\text{Var}(a - x_0 | x_t = x, s)$ can indeed be zero, even though $x_0$ is random. This is because the variance is conditional on $x_t = x$. For instance, in a 1-dimensional example where $a = 1$, $x_0$ follows a Gaussian distribution $N(0, 1)$, and we define $x_t = (1 - t) x_0 + t$, as in Rectified Flow. In this case, given $x_t = x$, we can deterministically solve $(a - x_0)$ because $x_0 = \frac{x_t - t}{1 - t}$ and $a$ is a constant 1. In fact, in this simple 1D example, the flow is exactly straight because $a$ is deterministic, which is what Proposition 3.2 is stating.
>
> - **Regarding Proposition 3.3**, the statement is accurate and correct. However, we will ensure that the statement is made more rigorously with a clear presentation of the assumptions and properties from earlier works.
>
> - The property mentioned is that $z_t$ and $x_t$ share the same law (i.e., for any $t$, their distributions are the same), which is formally stated as Theorem 3.3 in the Rectified Flow paper.
>
> - We respectfully note that we do not assume the conditional distributions of the actions are Gaussian. The action can follow any distribution, similar to the image distribution in image diffusion applications. Only the initial noise $x_0$ follows a Gaussian distribution. We would appreciate it if the reviewer could clarify what specific assumption or detail they are referring to regarding the Gaussian distribution.

---

> > ### Comment · Reviewer_FRk8 · 2024-08-10
> > **Thank you for the clarification**
> >
> > Thank you for the clarification regarding line 147.  On the subject of Proposition 3.3, I agree that the conclusion is correct, but I maintain that the proof as written is not rigorous and should be cleaned up.  I am happy to go along with the other reviewers on the subject of the experiments and raise my score accordingly due to the experimental results, but I still think that the theory is somewhat lacking.
> >
> > To clarify my point regarding the assumption of Gaussianity, note that equation (7), which you use to estimate the conditional variance is only consistent under the assumption that $a - x_0 - v$ is Gaussian.  While I believe that this may hold in practice due to universality properties of continuous stochastic processes, I think that this assumption should be more clearly stated and discussed.

---

> > > ### Author Response · Authors · 2024-08-14
> > > **Reply to Reviewer's Comment**
> > >
> > > We thank the reviewer for clarifying the question and raising the rating. Regarding the assumption that $a - x_0 - v$ is Gaussian, here is the explanation.
> > >
> > > For simplicity, we temporarily ignore the dependency on state $s_t$, then:
> > >
> > > $$
> > > L(\sigma) = \mathbb{E}\bigg[\int_{0}^1 \bigg(\frac{||(A - X_0) - v(X_t, t )||^2}{2\sigma(X_t, t)^2} + \log \sigma(X_t, t) \bigg) dt \bigg]
> > > $$
> > >
> > > First, assume $v(x_t, t)$ is learned well such that $v(x_t, t) = \mathbb{E}\big[ A - X_0 \mid X_t = x_t \big].$ This is guaranteed by the property of Rectified Flow.
> > >
> > > Next, take $\frac{dL}{d\sigma(x_t, t)}$ and set it to zero, we have:
> > >
> > > $$
> > > \frac{dL}{d\sigma(X_t, t)}|_{X_t = x_t} = \mathbb{E}\bigg[||(A - X_0)- v(X_t, t)||^2 \frac{-1}{\sigma(X_t, t)^3} + \frac{1}{\sigma(X_t, t)} \mid X_t = x_t\bigg] = 0
> > > $$
> > >
> > > Hence:
> > >
> > > $$
> > > \sigma(x_t, t)^2 = || (A - X_0) - v(X_t, t) \mid X_t = x_t ||^2 = \text{Var}(A - X_0 \mid X_t = x_t)
> > > $$
> > >
> > > As a result, note that though the particular objective of $L(\sigma)$ looks like the negative log likelihood of a Gaussian, **the closed-form of it indicates that the minimizer $\sigma^2$ corresponds exactly to the variance, no matter what true distribution it should be.** In other words, even if the $A - X_0 - v$ does not follow a Gaussian, minimizing Equation 7 (the objective of the $\sigma$ network) will still give the right variance estimation.

---

### Official Review · Reviewer_df3a · 2024-07-13

**Soundness:** 3
**Presentation:** 3
**Contribution:** 3
**Rating:** 7
**Confidence:** 4

**Summary:**

This paper proposes the AdaFlow algorithm for training fast and accurate imitation learning agents. AdaFlow learns a policy based on the flow generative model and makes use of the connection between the conditional variance of the training loss and the discretization error of the ODEs. In practice, it uses the estimated conditional variance to dynamically adapt the step size of the inference model. By doing so, it reduces inference time compared to diffusion models, and as the experiments show, learns a more accurate model.

**Strengths:**

* The paper shows an interesting idea - to use an estimation of the uncertainty in action prediction (the conditional variance) to dynamically adapt the step size of the flow model. If the model is certain in its action prediction, the step size will be large. Otherwise, the model performs many small prediction steps.
* AdaFlow speeds-up the inference step of flow-based models, which is an important factor for many real-time applications.
* The authors show the advantages of the proposed approach in various environments (4 tasks).
* The paper is well-written and easy to follow and understand.

**Weaknesses:**

Major:
* The main weakness of the paper is the lacking representation of the advantages/disadvantages of the main idea compared to Rectified Flow and Consistency models: The paper presents AdaFlow as the leading algorithm that reduces the inference time of diffusion policy, but it does not provide the full picture. As described in lines 199-206 (Discussion of AdaFlow and Rectified Flow), at inference time Rectified Flow allows for one-step action generation, which is either better than (for high variance actions) or equal (for low variance actions) to Adaflow. At training time, Rectified Flow involves a “reflow” stage after training the ODE. The reflow stage consists of generating pseudo-noise-data pairs and retraining the ODE. When comparing it to the training time of AdaFlow, AdaFlow also needs a two-stage training scheme for training the variance network (as described in lines 157-160), or a single, yet more involved training stage where the variance network is trained simultaneously with the ODE (which can also reduce performance). So both Rectified Flow and AdaFlow train two networks, the only difference is the generation of pseudo-noise-data pairs in Rectified Flow, which can be less time-consuming (or even negligible) compared to training the networks, depending on how many pairs are needed to get an accurate prediction.\
\
Overall, it seems that AdaFlow gains a reduction in training time (compared with Rectified Flow which generates pseudo-noise-data pairs) at the expense of longer inference time (due to additional steps for high variance actions). This means that it is not clear which algorithm is superior, and in general, it depends on the target task and dataset used.\
\
In addition, line 204 mentions that the reflow stage leads to inferior generation quality due to straightened ODEs, but the reason for this claim is unclear and lacks intuition, since training a network for variance prediction as done in AdaFlow can also lead to inaccuracies and suboptimality in some cases (also Tables 2 and 4 show that Rectified Flow performs better or equal to AdaFlow in about half of the environments examined).



* Comparison to the state of the art: The paper compares AdaFlow to the classical diffusion policy that uses DDIM and shows that AdaFlow achieves a higher success rate with less NFE. For example, Figure 3 shows that AdaFlow with NFE of 1.75 achieves comparable performance to a diffusion model with NFE of 100. To see the whole picture, it would be beneficial to compare AdaFlow to more advanced and faster solvers such as Dpm-solver and Dpm-solver++ (references 41 and 42 of the paper).

Minor mistakes:
* Figure 1 is not referenced in the paper and also the caption does not include an explanation of its subfigures.
* Line 175 - larger instead of large
* Table 5 - Maze 2 instead of Maze 1

**Questions:**

* In line 174 - should the inequality for epsilon be with $\sqrt{\eta}$ instead of $\eta$?
* Can the adaptive step be combined with a better ODE solver, such as RK4 (https://en.wikipedia.org/wiki/Runge%E2%80%93Kutta_methods)? if not, does AdaFlow perform better than RK4?
* Did you check other methods to estimate the uncertainty in action prediction (other than training a network that estimates the conditional variance)? For example, one can train an ensemble of agents and make use of the variance in their prediction to estimate uncertainty.
* Can the adaptive step size be implemented to generate images? If yes, what are the complications that may arise?

I am willing to raise my score if the authors address all of my concerns.

**Limitations:**

The authors did not sufficiently detail all the limitations of their algorithm - For example, not emphasizing the additional computation time and memory involved in training the variance network, and the additional inference steps compared to Rectified Flow.

---

> ### Author Rebuttal · Authors · 2024-08-07
>
> We thank Reviewer df3a for the thoughtful questions and suggestions.
>
> **1. What is the advantage of AdaFlow over Rectified Flow with reflow?**
>
> - **AdaFlow can learn action generation and variance estimation together, as shown in Appendix A.6**, where joint and separate training yield similar performance. We opted for separate training due to better efficiency and faster convergence, given the stationary objective of the variance network. Training the variance network is much quicker than training the original flow, adding minimal overhead to the 1-Rectified Flow. **In contrast, reflow and consistency distillation always require an expensive ODE distillation stage.**
>
> - Reflow is more complex and computationally expensive than AdaFlow. Reflow requires generating pseudo data by simulating the full trajectory (100 model inferences per pseudo sample) and needs a reflow dataset as large as the original. Then, it trains a new flow model with a reflow target, which can introduce errors. AdaFlow avoids this costly data generation. Training the variance estimation part separately (or jointly) involves only a few linear layers on the original rectified flow model. In practice, **on the maze task, reflow takes 7.5 times longer than AdaFlow in terms of wall clock time.**
>
> **2. Why does reflow lead to inferior generation quality with straightened ODE?**
>
> - Reflow estimates a **new mapping ODE** from noise to data, introducing errors. The first error comes from time-discretization during numerical simulation, and the second from training on pseudo data pairs. AdaFlow avoids these issues by using the original ODE.
>
> - In Tables 2 and 4, Reflow's rare outperformance of AdaFlow isn't due to variance prediction errors but a balance of efficiency and success rate. With more inference steps, AdaFlow outperforms Reflow, demonstrating a better ODE. The following tables compare Reflow and AdaFlow with different inference steps (NFE) on LIBERO tasks.
>
> 	|   | Task 1 | Task 2 | Task 3 | Task 4 | Task 5 | Task 6 | Average |
> 	| - | ------ | ------ | ------ | ------ | ------ | ------ | ------- |
> 	| Reflow  | 0.90 | 0.82  | 0.98   | 0.82   | 0.82   | 0.96   |  0.88   |
> 	| AdaFlow (NFE=1.27) | 0.98 | 0.80  | 0.98   | 0.82   | 0.90   | 0.96  |  0.91 |
> 	| AdaFlow (NFE=2.91) | 0.96 | 0.86  | 1.00   | 0.82   | 0.90   | 0.98  |  0.92 |
>
> **3. Comparison to DPM solver and DPM ++ solver**
>
> Thank you for suggesting a comparison with advanced solvers like DPM-solver and DPM-solver++. We conducted additional experiments, and as shown in the table below, AdaFlow achieves higher success rates even with fewer NFE across multiple LIBERO tasks.
>
> |   | NFE | Task 1 | Task 2 | Task 3 | Task 4 | Task 5 | Task 6 | Average |
> | - | ---   | ------ | ------ | ------ | ------ | ------ | ------ | ------- |
> | DPM     | 1    | 0.00 | 0.00  | 0.00   | 0.00   | 0.00   | 0.00  |  0.00 |
> | DPM     | 2    | 0.42 | 0.44  | 0.54   | 0.36   | 0.58   | 0.38  |  0.45 |
> | DPM     | 20   | 1.00 | 0.80  | 0.98   | 0.84   | 1.00   | 0.88  |  0.92 |
> | DPM++   | 1    | 0.00 | 0.00  | 0.00   | 0.00   | 0.00   | 0.00  |  0.00 |
> | DPM++   | 2    | 0.42 | 0.44  | 0.54   | 0.36   | 0.58   | 0.36  |  0.45 |
> | DPM++   | 20   | 1.00 | 0.80  | 0.98   | 0.84   | 1.00   | 0.92  |  0.92 |
> | AdaFlow | 1.27 | 0.98 | 0.80  | 0.98   | 0.82   | 0.90   | 0.96  |  0.91 |
>
> **4. Difference to RK45 solver**
>
> The RK45 solver adjusts step size for accuracy by using the difference between its 4th and 5th order estimates to calculate an error estimate, which then adjusts the next step size.
>
> AdaFlow and RK45 are orthogonal methods for step size estimation:
> - RK45 increases numerical simulation accuracy.
> - AdaFlow leverages the straightness of the learned flow.
>
> Thus, they can be used together. **The theoretical lowest NFE for RK45 is 6**, as it calculates 6 intermediate values per step, while AdaFlow needs close to one step.
>
> **5. Use other non-post-training methods to estimate the state variance**
>
> We explored estimating state variance without training a variance estimation network. We trained only the rectified flow model and estimated state variance by sampling. For a given state, we sampled $N_z$ noises and $N_t$ timesteps, and then estimate the variance of each state following Equation 6.
>
> $$\frac{1}{N_t} \frac{1}{N_z} \sum_t\sum_{z_0} \mathbb{E}[||y - z_0 - v(z_t, t; x)||^2]$$
>
> Here $y$ is derived using the Euler sampler, and $z_0$ is random Gaussian noise. This method estimates state variance accurately if both $N_z$ and $N_t$ is large enough. However, it involves sampling the full flow trajectory for many different noises. The figure below shows how many samples are needed to achieve a precise estimation of state variance: [Figure Link](https://anonymous.4open.science/r/adaflow_rebuttal-C0D6/variance_estimation.png). In general, an NFE > 10 is required to estimate the variance well. Compared to this method, AdaFlow is much faster for inference, only requiring one model inference.
>
> **6. Usage in generating images**
>
> Great point! We've also tried using AdaFlow for image generation. Applying the learned variance network at inference does speed up the process. However, the speedup is limited because the learned flow is not straight due to the high dimensionality and variability in image generation.
>
> In robotics tasks, many actions are deterministic given the states, unlike in image generation. For text-conditioned images, even detailed prompts can result in many possible images, making the generation trajectory non-deterministic. Thus, AdaFlow is better suited for robotics, where deterministic generations significantly reduce inference costs.
>
> **7. Typos**
> We will correct the typos mentioned by the reviewer and will reflect these changes in the revised version.
>
> Please let us know if you have any further questions or concerns. We hope that our rebuttal addresses all your points thoroughly, and we would greatly appreciate your reconsideration of the score. Thank you!

---

> > ### Comment · Reviewer_df3a · 2024-08-10
> > **Thank you for the answers**
> >
> > I thank the authors for their response. The authors address most of my concerns - The additional explanation regarding the training time of the variance network is important and should be added to the revised version. Also, the comparisons to DPM and DPM++  solvers are important and strengthen the paper's contribution.
> >
> >
> > Overall, considering the additional explanations and experiments provided in the rebuttal period and all reviewers' answers, I decided to raise my score to "accept".

---

### Decision · Program_Chairs · 2024-09-25

**Decision:**

Accept (poster)

**Comment:**

The work develops a flow simulation method in the context of imitation learning using a flow-based generative model. The method is based on the connection between variance and flow straightness, where the variance is estimated by an additional model which can be trained together with the flow model, which is then used to adjust discretization step size when interpreted as an indicator of flow curvature. All reviewers acknowledged the novel contribution and the significance to the field. Reviewers also raised a few questions and concerns, most of which have been reasonably addressed, while there also remain some unclear points, e.g., the applicability to high-dimensional problems (like image generation). Nevertheless, the paper has made a reasonable contribution, and is recommended for acceptance.